# Synthesis and Preliminary Biological Evaluation of Two Fluoroolefin Analogs of Largazole Inspired by the Structural Similarity of the Side Chain Unit in Psammaplin A

**DOI:** 10.3390/md17060333

**Published:** 2019-06-03

**Authors:** Bingbing Zhang, Guangsheng Shan, Yinying Zheng, Xiaolin Yu, Zhu-Wei Ruan, Yang Li, Xinsheng Lei

**Affiliations:** 1School of Pharmacy, Fudan University, 826 Zhangheng Road, Pudong Zone, Shanghai 201203, China; 14211030012@fudan.edu.cn (B.Z.); 12211030043@fudan.edu.cn (G.S.); 17211030013@fudan.edu.cn (Y.Z.); 13774294998@163.com (X.Y.); 13301030004@fudan.edu.cn (Z.-W.R.); 13211030007@fudan.edu.cn (Y.L.); 2College of Chemistry and Molecular Engineering, Zhengzhou University, Zhengzhou 450001, China; 3Key Laboratory of Synthetic Chemistry of Natural Substances, Shanghai Institute of Organic Chemistry, Chinese Academy of Sciences, Shanghai 200032, China

**Keywords:** marine natural product, Largazole, Psammaplin A, HDAC inhibitors, luoro olefin

## Abstract

Largazole, isolated from a marine *Cyanobacterium* of the genus *Symploca*, is a potent and selective Class I HDAC (histone deacetylation enzymes) inhibitor. This natural 16-membered macrocyclic depsipeptide features an interesting side chain unit, namely 3-hydroxy-7-mercaptohept-4-enoic acid, which occurs in many other natural sulfur-containing HDAC inhibitors. Notably, one similar fragment, where the amide moiety replaces the *trans* alkene moiety, appears in Psammaplin A, another marine natural product with potent HDAC inhibitory activities. Inspired by such a structural similarity, we hypothesized the fluoroolefin moiety would mimic both the alkene moiety in Largazole and the amide moiety in Psammaplin A, and thus designed and synthesized two novel fluoro olefin analogs of Largazole. The preliminary biological assays showed that the fluoro analogs possessed comparable Class I HDAC inhibitory effects, indicating that this kind of modification on the side chain of Largazole was tolerable.

## 1. Introduction

The regulation of histone-tailed lysine acetylation by histone acetyltransferases (HATs) and histone deacetylases (HDACs) plays a key role in the biochemistry of life process [1]. So far, at least 18 HDAC isoforms have been discovered, including Zn^2+^-dependent Class I (HDAC1, 2, 3, 8), Class II (class IIa: HDAC4, 5, 7, 9; class IIb: HDAC6, 10) and Class IV (HDAC11) [2]. Although the function of individual HDAC isoforms is not fully understood in cell and cancer biology, several pan-selective HDAC inhibitors, such as SAHA (Vorinostat), Belinostat, Panobinostat, Chidamide and Romidepsin (FK228), have been approved for the treatment of cutaneous T-cell lymphoma [2,3,4]. In principle, the class- and even isoform-specific HDAC inhibitors are hypothesized to be more beneficial in cancer therapy [5,6,7].

In 2008, Leusch’s group first discovered Largazole from a marine *Cyanobacterium* of the genus *Symploca* [8,9]. Largazole shows potent and highly selective inhibitory activities against Class I HDACs and displays superior anticancer properties, attracting great attention among many organic and medicinal chemists [10,11,12,13,14]. Largazole possesses a number of unusual structural features, including a 3-hydroxy-7-mercaptohept-4-enoic acid unit and a 16-membered macrocyclic depsipeptide moiety. Interestingly, this side chain unit also appears in several other natural HDAC inhibitors, including Romidepsin (FK228) [15], FR901375 [16,17], and Spiruchostatins (Figure 1) [18]. Psammaplin A, a member of a family of natural products isolated from some marine sponges, contains a similar side chain but with an amide bond instead of an olefin moiety (Figure 1) [19]. All these natural HDAC inhibitors can release a free sulfhydryl residue in vitro and in vivo through either reductive cleavage of the disulfide or hydrolysis of thioester, which is believed to exert inhibitory effects by coordinating with zinc cation of the HDACs (Figure 1) [9,10,11,12,13,14,19]. 

Up to now, great effort has been made to modify the 16-membered macrocyclic depsipeptide moiety of Largazole to search for more selective HDAC inhibitors [20], but modification on the common side chain unit usually leads to the loss of activities through changing the chain length and the stereo-configuration of the double bond and the secondary alcohol, as well as the Zn^2+^-binding group [21,22,23,24,25,26], possibly due to limitation of the narrow hydrophobic tunnel of HDACs [27]. Inspired by the structural similarity of the side chain unit of Largazole and Psammaplin, we believed that the unit may play a common role in the inhibition of HDACs. However, our previous work also demonstrated a complete loss of activity when the alkene moiety in Largazole was replaced with the amide moiety [28]. Comparing with *trans* olefin, *trans* fluoroolefin is usually viewed as a superior bioisostere of amide bond based on the steric demand, bond length and angle [29], and furthermore, the replacement of the olefin with fluoroolefin has little influence on the rigid conformation of Largazole. Herein, we report our work on synthesis and preliminary biological evaluation of two novel fluoro olefin analogs of Largazole.

## 2. Results and Discussion

### 2.1. Chemistry

The synthesis of two fluoroolefin analogs of Largazole was accomplished according to Scheme 1. The key 4-fluoro analog of chiral 3-hydroxy/amino-7-mercaptohept-4-enoic acid was prepared from acrolein in four steps. Conjugate addition of triphenylmethanethiol (**1**) to acrolein **2** gave **3**, which was directly used in the following Wittig-Type reaction [30] to give rise to **4** in 97% yield with *Z*- and *E*-isomer ratio of 3/1. Although the isomers could be carefully separated by column chromatography, most of the undesired *E*-isomer was found to be isomerized to the *Z*-isomer during the reduction with DIBALH in anhydrous THF. As a result, a mixture of *Z*- and *E*-isomer **4** was subjected to the reduction condition, giving the fluoro unsaturated aldehyde **5** in 86% yield. The asymmetric aldol condensation of **5** with acetyl Nagao’s auxiliary afforded 4-fluoro derivatives of chiral 3-hydroxy-7-mercaptohept-4-enoic acid (**8** and **9**) under Xie’s conditions [31,32] at a low temperature (−90 °C). The two diastereomers with 2:1 ratio as determined by ^19^F-NMR, were readily separated by silica gel column chromatography. The major chiral alcohol **8** was assigned to the desired *S*-configuration, while the minor alcohol **9** was designated as the *R*-configuration, according to the known reaction mechanism [32].

Compound **8** could be directly used for the next aminolysis as Nagao [31] and Xie [32] reported. Aminolysis of **8** with the thiazole-thiazoline fragment (**10**) [28] under the basic condition provided **11** in a good yield (70%). Then, esterification of the secondary alcohol in **11** with Fmoc-l-Valine (**12a**) or Fmoc-l-Phenylalanine (**12a**) pre-activated by 2,4,6-Cl_3_-C_6_H_2_COCl in the presence of DMAP and DIEPA [28], afforded **13a** and **13b** (yields: 65% and 60%, respectively). The structures of **13a** and **13b** were subsequently confirmed by ^1^H, ^13^C, COSY, HMQC and HMBC NMR spectra. Careful methyl ester hydrolysis of **13a** or **13b** in LiOH solution (THF/H_2_O) and removal of Fmoc group with Et_2_NH in CH_2_Cl_2_ resulted in the linear depsipeptide exposed at the N- and C-terminus, and after examination of several conditions, the system of HATU/HOAT/DIPEA in anhydrous CH_2_Cl_2_ solution (the concentration of cyclization precursor at about 0.001 M) was used for the macrolactamization, giving **14a** and **14b** (yields in three steps: 31% and 22%, respectively). Finally, deprotection of Trt group in **14a** and **14b**, using (i-Pr)_3_SiH (TIPS) and CF_3_CO_2_H in anhydrous CH_2_Cl_2_ solution, proceeded smoothly to afford the free thiol **15a** and **15b** (yields: 67% and 59%, respectively). The subsequent acylation with *n*-C_7_H_15_COCl under the standard condition led to the final fluoro analogs **16a** and **16b** (yields: 61% and 66%, respectively).

### 2.2. Biology

In vitro enzymatic assays on Largazole, Largazole free thiol and the related compounds (**16a**, **15a** and **15b**) were performed according to the known method [33] with SAHA (Vorinostat) as a reference. The inhibitory potencies against human HDACs (Class I HDACs: 1, 2, 3 and 8; one representative of Class II HDACs: HDAC 6) were evaluated, and the results are shown in Table 1 and Figure 2.

As expected, Largazole displayed relatively weaker HDACs inhibitory activities (IC_50_: from 120 to *ca.* 5000 nM) in vitro enzymatic assays. In contrast, Largazole thiol, the possibly activated species, displayed potent inhibitory activities on Class I HDACs with IC_50_ values in the nanomolar range (IC_50_ values of HDAC 1, 2, 3 and 8: 2.0, 9.5, 14.5 and 3.8 nM, respectively), and exhibited up to 61-fold selectivity over Class II HDACs exemplified by the IC_50_ value ratio of HDAC 1 versus HDAC 6. Most of these results were comparable with those reported [8,9,10,11,20,23,34] except HDAC 8 activity (3.8 nM verus 102 nM reported by Leusch). Despite that, the above results confirmed that Largazole thiol was a potent and selective Class I HDAC inhibitor, compared to SAHA.

Similar to Largazole, the fluoro ester **16a**, also displayed relatively weaker HDACs inhibitory activities, and the fluoro thiol **15a** exhibited highly potent inhibitory activities on Class I HDACs (IC_50_ values of HDAC 1, 2, 3 and 8: 4.4, 21.0, 39.2 and 8.3 nM, respectively) and almost the same selectivity over Class II HDACs (about 68-fold selectivity towards HDAC 1 over HDAC 6). These results indicated that the *trans* alkene moiety in Largazole could be replaced with *trans* fluoroolefin, which was significantly different from the previous results, because the replacement of *trans* olefin with an amide bond or aryl ring could lead to a complete loss of activities [14,28]. Furthermore, in contrast to those modifications on the chain length, the stereo-configuration of the double bond or the secondary alcohol of the linker [21,22,23,24,25,26], our modification was tolerable.

Compared with **15a**, the other fluoro thiol (**15b**), with a benzyl group instead of an *i*-propyl group at C2-position of the macrocycle, also exhibited highly potent and selective inhibitory activities on Class I HDACs (IC_50_ values of HDAC 1, 2, 3, 8 and 6: 4.2, 16.2, 37.1, 8.3 and 338 nM, respectively). These results indicated that the modification at C2-position could be permitted to some extent with no loss of the activity and selectivity, which was also consistent with the results through the same modification on Laragzole thiol [25].

In the in vitro cell assays, considering the poor cell permeability, Largazole and its analog (**16a**), instead of their corresponding free thiols, were tested against several cell lines, and the results are shown in Table 2 and Figure 3. Largazole exhibited evident growth inhibition on A549, HCT116, MDA-MB-231, and SK-OV-3 cells with IC_50_ values of 0.46, 0.184, 1.37 and 0.034 µM, respectively. Comparing our data of HCT116 with Luesch’s [9], we found our data was about four-fold less potent than the reported data (0.184 µM versus 0.044 µM), indicating our assays were robust. However, in contrast to nanomolar-range potencies in the enzymatic assays, sub-micromolar-range potencies were observed in the cellular assay. We reasoned that Largazole might extracellularly release the free thiol to some extent in the cellular assays and then result in the diminished potencies due to the poor cell permeability of the free thiol.

As expected, the fluoro olefin analog (**16a**) gave comparable activities in the A549, HCT116, MDA-MB-231 cellular assay (IC_50_, A549: 0.52 µM, HCT116: 0.81 µM; MDA-MB-231: 3.85 µM) except SK-OV-3 (IC_50_, 0.25 µM). We reasoned that there may be more extracellular release of its free thiol in the case of SK-OV-3. Taken together, based on both enzymatic and cellular assays, we demonstrated that fluorine could be introduced into the alkene moiety of the Largazole with little sacrifice of the activities and selectivity. The kind of modification might be feasible for this family of the natural products (Figure 1) sharing the common side chain, and further study is under way.

## 3. Materials and Methods 

### 3.1. Chemistry

The chemicals and reagents were purchased from Acros, Alfa Aesar, and National Chemical Reagent Group Co. Ltd., Shanghai, China, and used without further purification. Anhydrous solvents (THF, MeOH, DMF, CH_2_Cl_2_, and CH_3_CN) used in the reactions were dried and freshly distilled before use. All the reactions were carried out under Ar atmosphere, otherwise stated else. The progress of the reactions was monitored by TLC (silica-coated glass plates) and visualized under UV light, and by using iodine or phosphomolybdic acid. Melting points were measured on an SGW X-4 microscopy melting point apparatus without correction. ^1^H-NMR and ^13^C-NMR spectra were recorded either on a 400 MHz Varian Instrument at 25 °C or 600 MHz Bruker Instrument at 25 °C, using TMS as an internal standard, respectively. Multiplicity is tabulated as s for singlet, d for doublet, dd for doublet of doublet, t for triplet, and m for multiplet. The original spectra of the relative compounds could be found in Appendix A. HRMS spectra were recorded on Finnigan-Mat-95 mass spectrometer, equipped with an ESI source. Experimental procedures for the preparation of compound **10** and Largazole are available in the reported literature [28].


*3-(Tritylthio)propanal (*
**3**
*)*


To a solution of triphenylmethyl mercaptan (1.38 g, 5.0 mmol, 1.0 equiv.) in CH_2_Cl_2_ (DCM, 30 mL) acrolein (0.393 g, 7.0 mmol, 1.4 equiv.) and triethylamine (0.98 mL, 7.0 mmol, 1.4 equiv.) was added, and the resulting mixture was stirred at room temperature for 1 h. Concentration in vacuo left aldehyde **3** as a white foamy solid. *R*_f_ = 0.13 [petroleum ether (PE)/ethyl acetate (EA) 40/1], ^1^H-NMR (400 MHz, CDCl_3_): δ 9.56 (brs, 1H), 7.23–7.43 (m, 15H), 2.47 (t, *J* = 7.0 Hz, 2H), 2.37 (t, *J* = 6.7 Hz, 2H).


*Ethyl 2-fluoro-5-(tritylthio)pent-2-enoate (*
**4**
*)*


To an anhydrous THF solution (30 mL) of PPh_3_ (3.15 g, 12.0 mmol, 4.0 equiv) and ethyl bromodifluoroacetate (0.83 mL, 6.0 mmol, 2.0 equiv), 1.0 M Et_2_Zn in hexane (12 mL, 12.0 mmol, 4.0 equiv) was rapidly added under argon. The mixture was stirred for 10 min (until the internal temperature returned to rt), then the aldehyde **3** (0.997 g, 3.0 mmol) was rapidly added. The mixture was stirred overnight. The resulting solution was then quenched with EtOH (15 mL), stirred for 15 min, and concentrated under reduced pressure. The residue was taken up in Et_2_O (100 mL) and filtered through Celite and then it was chromatographed (silica gel, PE/EA 40/1) to afford the ɑ-fluoroacrylate **4**, with *E*-isomer (0.325 g, 26%) and *Z*-isomer (0.730 g, 58%), respectively, or afford **4** with a mixture of them (*Z*/*E* 3/1, determined by ^19^F NMR) in 97% yield.

For *Z*-**4**: *R*_f_ = 0.35 (PE/EA 40/1). ^1^H-NMR (400 MHz, CDCl_3_): δ 7.40 (m, 6H), 7.25 (m, 6H), 7.18 (m, 3H), 5.75 (dt, *J* = 20.9, 7.9 Hz, 1H), 4.23 (q, *J* = 7.1 Hz, 2H), 2.56 (m, 2H), 2.26 (t, *J* = 7.2 Hz, 2H), 1.29 (t, *J* = 7.1 Hz, 3H). ^13^C-NMR (150 MHz, CDCl_3_): δ 160.58 (d, *^2^J*_C-F_ = 34.5 Hz), 147.91 (d, ^1^*J*_C-F_ = 252 Hz), 144.65, 129.48, 127.81, 126.60, 121.24 (d, *^2^J*_C-F_ = 19.6 Hz), 66.71, 61.28, 31.30, 24.55 (d, ^3^*J*_C-F_ = 5.1 Hz), 14.00. ^19^F-NMR (376 MHz, CDCl_3_): δ –121.29 (d, *J* = 22.6 Hz). ESI-MS (*m/z*): 443.6 [M + Na]^+^. HRMS-ESI (*m/z*): [M + Na]^+^ calcd. for C_26_H_25_FO_2_SNa: 443.1452, found: 443.1452.

For *E*-**4**: *R*_f_ = 0.27 (PE/EA 40/1). ^1^H-NMR (400 MHz, CDCl_3_): δ 7.41 (m, 6H), 7.28 (m, 6H), 7.22 (m, 3H), 5.97 (dt, *J* = 32.8, 7.2 Hz, 1H), 4.25 (q, *J* = 7.1 Hz, 2H), 2.25 (m, 4H), 1.31 (t, *J* = 7.1 Hz, 3H). ^13^C-NMR (150 MHz, CDCl_3_): δ 155.83 (d, *^2^J*_C-F_ = 34.5 Hz), 143.62 (d, *^1^J*_C-F_ = 256.5 Hz), 139.93, 124.82, 123.19, 121.99, 113.57 (d, *^2^J*_C-F_ = 11.4 Hz), 62.15, 56.85, 25.79, 18.90, 9.39. ^19^F-NMR (376 MHz, CDCl_3_): δ –128.95 (d, *J* = 32.8 Hz). ESI-MS (*m/z*): 443.6 [M + Na]^+^. HRMS-ESI (*m/z*): [M + Na]^+^ calcd. for C_26_H_25_FO_2_SNa: 443.1452, found: 443.1454.


*(Z)-2-Fluoro-5-(tritylthio)pent-2-enal (*
**5**
*)*


To an anhydrous toluene solution (100 mL) of a mixture of *Z*-**4** and ***E*-4** prepared above (4.03 g, 9.58 mmol) 1.5 M DIBALH in toluene (22.4 mL, 33.5 mmol) was added dropwise under argon at −78 °C over 0.5 h. The mixture was stirred at −78 °C for 1 h, and the resulting solution was then quenched with MeOH (50 mL), and warmed to rt. After addition of the saturated aqueous solution of Rochelle salt, the mixture was stirred for overnight, and then the organic phase was separated. The aqueous phase was extracted with EA (100 mL × 2), and the combined organic extracts were dried, filtered, and concentrated in vacuo, and then it was chromatographed (silica gel, PE/DCM/EA 35/5/1) to afford the aldehyde **5** as a white foamy solid (3.10 g, 86%). *R*_f_ = 0.35 (PE/DCM/EA 35/5/1). ^1^H-NMR (400 MHz, CDCl_3_): δ 9.13 (d, *J* = 18.2 Hz, 1H), 7.43 (m, 6H), 7.27 (m, 9H), 5.75 (dt, *J* = 32.1, 7.2 Hz, 1H), 2.35 (m, 4H). ^13^C-NMR (100 MHz, CDCl_3_): δ 183.49 (d, *^2^J*_C-F_ = 24.9 Hz), 156.48 (d, *^1^J*_C-F_ = 261 Hz), 146.88, 144.53, 129.54, 128.71 (d, *^2^J*_C-F_ = 10.2 Hz), 127.99, 127.92, 127.29, 126.86, 67.09, 30.22, 24.01. ^19^F-NMR (376 MHz, CDCl_3_): δ −132.15 (dd, *J* = 32.1, 18.2 Hz). ESI-MS (*m/z*): 399.6 [M + Na]^+^. HRMS-ESI (*m/z*): [M + Na]^+^ calcd. for C_24_H_21_FOSNa: 399.1189, found: 399.1192.


*(S/R,Z)-1-((R)-4-Benzyl-2-thioxothiazolidin-3-yl)-4-fluoro-3-hydroxy-7-(tritylthio)hept-4-en-1-one (*
**8/9**
*)*


**6** (1.12 g, 4.46 mmol) was dissolved in dry DCM (50 mL) and cooled to 0 °C. TiCl_4_ (0.82 mL, 7.43 mmol) was added dropwise. After stirring for 30 min, the resulting yellow suspension was cooled to −40 °C. *i*-Pr_2_NEt (1.32 mL, 7.97 mmol) was then added dropwise, and the reaction mixture was stirred for 2 h at this temperature. The resulting solution was then cooled to −90 °C. **5** (1.44 g, 3.71 mmol) was dissolved in dry DCM (20 mL) and added dropwise to the reaction mixture. The reaction mixture was stirred at −90 °C for 3 h and quenched by the addition of a saturated aqueous solution of NH_4_Cl. The reaction mixture was then allowed to warm to room temperature and extracted three times with DCM (20 mL × 3). The combined organic layers were washed with brine, dried over Na_2_SO_4_, filtered, and the solvent evaporated under reduced pressure. The resulting crude was purified by flash chromatography on silica gel, eluting with PE/EA (8/1) to afford **8** (1.33 g, 59%) and **9** (0.656 g, 29%) as yellow oils, respectively.

For **8**: *R*_f_ = 0.19 (PE/EA 4/1). [α]^20^_D_: −78.0 (*c* 3.7, CHCl_3_). ^1^H-NMR (400 MHz, CDCl_3_): δ 7.29 (m, 20H), 5.33 (m, 1H), 4.86 (dd, *J* = 36.8, 7.1 Hz, 1H), 4.63 (brs, 1H), 3.67 (m, 1H), 3.47 (dd, *J* = 17.9, 8.8 Hz, 1H), 3.36 (dd, *J* = 11.2, 7.2 Hz, 1H), 3.21 (m, 1H), 3.03 (m, 1H), 2.95 (d, *J* = 4.1 Hz, 1H), 2.87 (d, *J* = 11.6 Hz, 1H), 2.19 (m, 4H). ^13^C-NMR (150 MHz, CDCl_3_): δ 201.27, 172.02, 158.60 (d, *^1^J*_C-F_ = 257.2 Hz), 144.87, 136.32, 129.60, 129.45, 128.97, 127.88, 127.34, 126.63, 104.96 (d, *^2^J*_C-F_ = 12.9 Hz), 68.32, 66.63, 42.81, 36.79, 32.18, 31.52, 22.72. ^19^F-NMR (376 MHz, CDCl_3_): δ –129.15 (dd, *J* = 32.1, 18.2 Hz). ESI-MS (*m/z*): 650.4 [M + Na]^+^. HRMS-ESI (*m/z*): [M + Na]^+^ calcd. for C_36_H_34_FNO_2_S_3_Na: 650.1628, found: 650.1621. 

For **9**: *R*_f_ = 0.30 (PE/EA 4/1). [α]^20^_D_: −65.4 (*c* 0.30, CHCl_3_). ^1^H-NMR (400 MHz, CDCl_3_): δ 7.30 (m, 20H), 5.37 (m, 1H), 4.85 (dt, *J* = 37.2, 7.0 Hz, 1H), 4.56 (brs, 1H), 3.75 (dd, *J* = 17.5, 9.2 Hz, 1H), 3.41 (m, 2H), 3.21 (m, 2H), 3.03 (dd, *J* = 13.2, 10.4 Hz, 1H), 2.89 (d, *J* = 11.6 Hz, 1H), 2.19 (m, 4H). ^13^C-NMR (150 MHz, CDCl_3_): δ 201.27, 172.02, 158.60 (d, *^1^J*_C-F_ = 257.2 Hz), 144.87, 136.32, 129.60, 129.45, 128.97, 127.88, 127.34, 126.63, 104.96 (d, *^2^J*_C-F_ = 12.9 Hz), 68.32, 66.63, 42.81, 36.79, 32.18, 31.52, 22.72. ESI-MS (*m/z*): 650.4 [M + Na]^+^. HRMS-ESI (*m/z*): [M + Na]^+^ calcd. for C_36_H_34_FNO_2_S_3_Na: 650.1628, found: 650.1621.


*(R)-Methyl 2′-(((S/R,Z)-4-fluoro-3-hydroxy-7-(tritylthio)hept-4-enamido)methyl)-4-methyl-4,5-dihydro-[2,4′-bithiazole]-4-carboxylate (31S/31R) (*
**11**
*)*


To a stirring solution of **10** (0.459 g, 1.28 mmol) and DMAP (0.408 g, 3.34 mmol) in anhydrous DCM (20 mL) a solution of **8** (0.820 g, 1.28 mmol) in DCM (20 mL) was added dropwise at rt, and the resultant solution was stirred for another 2 h. The solution was then quenched with NH_4_Cl saturated solution (20 mL) and separated, and the aqueous phase was extracted with DCM (30 mL × 3). The combined organic phase was washed successively with H_2_O, brine, dried with Na_2_SO_4_ and filtered. After removal of the solvent, the residue was purified by flash chromatography on silica gel with DCM/EA (2/1) to yield **11** (0.600 g, 70%) as a foamy solid.

*R*_f_ = 0.26 (PE/EA 1/1). [α]^20^_D_: −32.2 (*c* 3.6, CHCl_3_). ^1^H-NMR (400 MHz, CDCl_3_): δ 7.89 (s, 1H), 4.79 (m, 2H), 4.68 (m, 2H), 4.49 (brs, 1H), 3.87 (d, *J* = 11.4 Hz, 1H), 3.79 (s, 3H), 3.27 (d, *J* = 11.4 Hz, 1H), 2.56 (m, 2H), 2.15 (m, 5H), 1.63 (s, 3H). ^13^C-NMR (100 MHz, CDCl_3_): δ 173.62, 171.49, 167.63, 162.78, 158.92 (d, *^1^J*_C-F_ = 257.4 Hz), 148.20, 144.83, 129.58, 127.91, 126.67, 122.56, 104.67 (d, *^2^J*_C-F_ = 12.3 Hz), 84.49, 66.97 (d, *^2^J*_C-F_ = 33 Hz), 66.62, 53.01, 41.54, 40.80, 39.72, 31.49, 24.02, 22.70 (d, *^3^J*_C-F_ = 3.5 Hz). ^19^F-NMR (376 MHz, CDCl_3_): δ −124.95 (dd, *J* = 36.2, 20.3 Hz). ESI-MS (*m/z*): 712.4 [M + Na]^+^. HRMS-ESI (*m/z*): [M + Na]^+^ calcd. for C_36_H_36_FN_3_O_4_S_3_Na: 712.1744, found: 712.1748.


*(R)-Methyl 2′-((5S/R,8S)-1-(9H-fluoren-9-yl)-8-((Z)-1-fluoro-4-(tritylthio)but-1-en-1-yl)-5-isopropyl-3,6,10-trioxo-2,7-dioxa-4,11-diazadodecan-12-yl)-4-methyl-4,5-dihydro-[2,4′-bithiazole]-4-carboxylate (*
**13a**
*)*


To a solution of Fmoc-Val-OH (0.165 g, 0.49 mmol) and DMAP (0.005 g, 0.041 mmol) in DCM (15 mL) 2,4,6-trichlorobenzoyl chloride (0.089 mL, 0.568 mmol) was added at 0 °C, and then the solution continued to stirred for 1 h. Then. a solution of **11** and DIPEA (0.11 mL, 0.65 mmol) in DCM (10 mL) was added to the stirring mixture. The reaction was allowed to warm to rt and stirred for 1 h. This solution was quenched with a NH_4_Cl aqueous solution and separated. The aqueous phase was extracted with DCM (30 mL × 3) and the combined organic phase was washed successively with H_2_O, brine, dried with Na_2_SO_4_ and filtered. After the removal of the solvent, the resulting crude was purified by flash chromatography on silica gel, eluting with PE/EA (5/4) to afford **13a** (0.266 g, 65%) as a white foamy solid. *R*_f_ = 0.26 (PE/EA 1/1). [α]^20^_D_: −16.4 (*c* 0.34, CHCl_3_). ^1^H-NMR (600 MHz, CDCl_3_): δ 7.89 (s, 1H), 7.75 (d, *J* = 7.2 Hz, 2H), 7.57 (d, *J* = 3.6 Hz, 2H), 7.38 (m, 8H), 7.34–7.24 (m, 8H), 7.20 (t, *J* = 7.1 Hz, 3H), 6.64 (brs, 1H), 5.67–5.79 (m, 1H), 5.25 (brs, 1H), 4.92 (dt, *J* = 35.9, 7.0 Hz, 1H), 4.68 (m, 2H), 4.37 (m, 2H),4.20 (t, *J* = 6.9 Hz, 1H), 4.16 (dd, *J* = 7.6, 5.8 Hz, 1H), 3.85 (d, *J* = 11.3 Hz, 1H), 3.78 (s, 3H), 3.24 (d, *J* = 11.3 Hz, 1H), 2.70 (m, 2H), 2.01–2.24 (m, 5H), 1.63 (s, 3H), 0.88 (t, *J* = 6.5 Hz, 3H), 0.83 (d, *J* = 6.6 Hz, 3H). ^13^C-NMR (150 MHz, CDCl_3_): δ 173.62, 170.92, 168.24, 167.81, 162.77, 156.29, 153.75 (d, *^1^J*_C-F_ = 257.4 Hz), 148.35, 144.78, 143.79 (d, *J*_C-F_ = 19.3 Hz), 141.33, 129.58, 127.91, 127.82 (d, *J*_C-F_ = 24.5 Hz), 127.12, 126.68, 125.08, 122.33, 120.00, 109.63 (d, *^2^J*_C-F_ = 12.0 Hz), 84.56, 69.62 (d, *^3^J*_C-F_ = 28.5 Hz), 67.08, 66.71, 59.17, 52.91, 47.19, 41.53, 41.09, 38.02, 31.59, 31.10 (d, *J* = 10.2 Hz), 23.98, 22.79, 22.66, 18.81, 17.71. ^19^F NMR (376 MHz, CDCl_3_): δ −125.77 (dd, *J* = 35.4, 21.1 Hz). ESI-MS (*m/z*): 1011.2 [M + H]^+^. HRMS-ESI (*m/z*): [M+Na]^+^ calcd. for C_56_H_55_FN_4_O_7_S_3_Na: 1033.3109, found: 1033.3102.

For **13b**: 60%, *R*_f_ = 0.26 (PE/ EA 1/1). [α]^20^_D_: −4.7 (c = 0.097, CHCl_3_). ^1^H-NMR (600 MHz, CDCl_3_) δ 7.89 (s, 1H), 7.76 (d, *J* = 7.5 Hz, 2H), 7.53 (t, *J* = 7.2 Hz, 2H), 7.39 (d, *J* = 7.4 Hz, 8H), 7.33–7.25 (m, 9H), 7.19 (dd, *J* = 16.7, 7.7 Hz, 6H), 7.04 (d, *J* = 6.9 Hz, 2H), 6.60 (s, 1H), 5.77–5.57 (m, 1H), 5.26 (d, *J* = 7.5 Hz, 1H), 4.81 (dt, *J* = 36.0, 7.0 Hz, 1H), 4.70–4.61 (m, 2H), 4.52 (dd, *J* = 12.9, 6.3 Hz, 1H), 4.45–4.27 (m, 2H), 4.18 (t, *J* = 6.8 Hz, 1H), 3.86 (d, *J* = 11.3 Hz, 1H), 3.78 (s, 3H), 3.25 (d, *J* = 11.3 Hz, 1H), 3.03 (m, 2H), 2.66 (m, 2H), 2.18 (t, *J* = 6.5 Hz, 3H), 2.15–2.07 (m, 2H), 1.63 (s, 4H). ^13^C-NMR (150 MHz, CDCl_3_) δ 172.56, 169.70, 167.54, 167.03, 162.15, 154.99 (d, *J* = 11.5 Hz), 154.46 (d, *J* = 14.7 Hz), 152.56 (d, *J* = 74.3 Hz), 147.53 (d, *J* = 54.5 Hz), 144.14, 143.09 (d, *J* = 14.9 Hz), 140.69, 134.75, 128.94, 128.73, 127.94, 127.29, 127.11, 126.50 (d, *J* = 7.5 Hz), 126.07, 124.44 (d, *J* = 7.6 Hz), 121.71, 119.37, 109.04 (d, *J* = 12.3 Hz), 83.92, 69.32 (d, *J* = 29.3 Hz), 66.25 (d, *J* = 46.7 Hz), 54.26, 52.28, 46.50, 40.89, 40.41, 37.41, 37.18, 30.43, 23.36, 22.31. ^19^F-NMR (376 MHz, CDCl_3_): δ −126.27 (dd, *J* = 35.4, 21.1 Hz). ESI-MS (*m/z*): 1059.2 [M + H]^+^. HRMS-ESI (*m/z*): [M + Na]^+^ calcd. for C_60_H_55_FN_4_O_7_S_3_Na:1081.3109, found: 1081.3126.


*(5R,8S,11S/R)-11-((Z)-1-Fluoro-4-(tritylthio)but-1-en-1-yl)-8-isopropyl-5-methyl-10-oxa-3,17-dithia-7,14,19,20-tetraazatricyclo[14.2.1.12,5]icosa-1(18),2(20),16(19)-triene-6,9,13-trione (*
**14a**
*)*


To a solution of **13a** (0.420 g, 0.42 mmol) in THF/H_2_O (4/1, v/v, 20 mL) LiOH∙H_2_O (0.027 g, 0.63 mmol) was added at 0 °C, and the reaction mixture was stirred for 3 h at that temperature. After the hydrolysis was completed, the reaction mixture was quenched with 1.0 M HCl aqueous solution (1.0 mL), and then extracted with DCM (20 mL × 3). The combined organic layers were washed successively with water, brine, dried with Na_2_SO_4_ and filtered. After evaporation, the residue was purified by flash chromatography on silica gel, eluting with DCM/MeOH (40/1 to 10/1) to afford the resultant acid (0.251 g) as a white foamy solid.

The resultant acid (0.251 g) was dissolved in dry DCM (20 mL), and Et_2_NH (1.0 mL) was added at rt. After stirring for 3 h and removal of the solvent, toluene (10 mL) was added and evaporated under reduced pressure, and this operation was repeated three times to remove the remaining Et_2_NH, affording the crude product. 

Then, the crude amino acid was dissolved in dry DCM (100 mL), and the solution was added dropwise to a stirring solution of HATU (0.380 g, 1.0 mmol), HOAT (0.136 g, 1.0 mmol), and DIPEA (0.33 mL, 2.0 mmol) in DCM (300 mL) at rt. The reaction mixture was stirred at rt for 12 h. The solution was concentrated under reduced pressure, and this resultant solution (100 mL) was washed with saturated NH_4_Cl solution, water, and brine, successively, dried over Na_2_SO_4_, and filtered. After removal of the solvent, the residue was purified by flash chromatography on silica gel, eluting with PE/EA/MeOH (20/20/8) to afford **14a** (31%, over three steps) as a white foamy solid. *R*_f_ = 0.15 (PE/EA 2/3). [α]^20^_D_ −14.3 (*c* 0.4, CHCl_3_). ^1^H-NMR (600 MHz, CDCl_3_): δ 7.76 (s, 1H), 7.29 (m, 15H), 7.13 (d, *J* = 9.2 Hz, 1H), 6.41 (d, *J* = 7.6 Hz, 1H), 5.60 (dd, *J* = 20.2, 10.5 Hz, 1H), 5.29 (dd, *J* = 17.5, 9.7 Hz, 1H), 5.00 (m, 1H), 4.62 (d, *J* = 7.6 Hz, 1H), 4.20 (d, *J* = 18.1 Hz, 1H), 4.04 (d, *J* = 11.0 Hz, 1H), 3.28 (d, *J* = 11.3 Hz, 1H), 3.12 (m, 1H), 2.70 (d, *J* = 16.6 Hz, 1H), 2.15 (m, 5H), 1.85 (s, 3H), 0.68 (d, *J* = 6.4 Hz, 3H), 0.49 (d, *J* = 6.6 Hz, 3H). ^13^C-NMR (150 MHz, CHCl_3_): δ 173.60, 168.94, 168.76, 167.89, 164.58, 155.25 (d, *^1^J*_C-F_ = 256 Hz), 147.49, 129.58, 127.91, 126.65, 124.33, 109.26 (d, *^2^J*_C-F_ = 12.4 Hz), 84.39, 69.95 (d, *^3^J*_C-F_ = 29.6 Hz), 57.58, 43.32, 41.13, 38.63, 37.49, 34.23, 31.12, 29.72, 24.13, 22.88, 18.84, 16.57, 14.22. ^19^F NMR (376 MHz, CDCl_3_): δ –124.95 (dd, *J* = 36.2, 20.3 Hz). ESI-MS (*m/z*): 779.4 [M + Na]^+^. HRMS-ESI (*m/z*): [M + Na]^+^ calcd. for C_40_H_41_FN_4_O_4_S_3_Na: 779.2166, found: 779.2171.

For **14b**: 22%, *R*_f_ = 0.15 (PE/EA 2/3). [α]^20^_D_: + 28.6 (*c* 0.021, CHCl_3_), ^1^H-NMR (600 MHz, CDCl_3_) δ 7.61 (s, 1H), 7.38 (d, *J* = 7.6 Hz, 6H), 7.27 (dd, *J* = 11.0, 4.3 Hz, 6H), 7.20 (t, *J* = 7.3 Hz, 3H), 7.13 (d, *J* = 7.7 Hz, 1H), 6.94–6.70 (m, 5H), 6.15 (d, *J* = 8.3 Hz, 1H), 5.71 (m, *J* = 18.9, 10.1, 1.9 Hz, 1H), 5.05 (d, 12 Hz, 1H), 5.00 (dt, 36 Hz, 7.5Hz, 1H), 4.88 (m, 1H), 4.17 (dd, *J* = 17.4, 2.8 Hz, 1H), 4.08 (d, *J* = 11.3 Hz, 1H), 3.25 (d, *J* = 11.3 Hz, 1H), 3.20 (dd, *J* = 14.0, 23 Hz, 1H), 3.06 (dd, *J* = 14.0, 5.9 Hz, 1H), 2.99 (dd, *J* = 16.4, 10.2 Hz, 1H), 2.62 (dd, *J* = 16.4, 2.1 Hz, 1H), 2.17 (dt, *J* = 11.0, 4.0 Hz, 2H), 2.10 (dt, *J* = 14.4, 7.2 Hz, 2H), 1.79 (s, 3H). ^13^C-NMR (150 MHz, CDCl_3_): δ 173.67, 168.89, 168.16, 166.97, 163.73, 155.33, 153.63, 147.26, 144.78, 135.11, 129.60 (d, *J* = 8.8 Hz), 127.88 (d, *J* = 7.6 Hz), 126.65, 125.93, 123.68, 109.17 (d, *J* = 12.5 Hz), 84.19, 70.18 (d, *J* = 30.9 Hz), 66.61, 54.19, 42.46, 40.98, 37.70, 37.46, 31.12, 24.98, 22.88 (d, *J* = 3.8 Hz). ^19^F-NMR (376 MHz, CDCl_3_): δ –124.43 (dd, *J* = 36.2, 20.3 Hz). ESI-MS (*m/z*): 827.2 [M + Na]^+^. HRMS-ESI (*m/z*): [M + Na]^+^ calcd. for C_44_H_41_FN_4_O_4_S_3_Na: 827.2166, found: 827.2166.


*(5R,8S,11S/R)-11-((Z)-1-Fluoro-4-mercaptobut-1-en-1-yl)-8-isopropyl-5-methyl-10-oxa-3,17-dithia-7,14,19,20-tetraazatricyclo[14.2.1.12,5]icosa-1(18),2(20),16(19)-triene-6,9,13-trione (*
**15a**
*)*


**14a** (50 mg, 0.066 mmol) was dissolved in dry DCM (15 mL) and cooled to 0 °C. The mixture was successively treated with Et_3_SiH (27 µL, 0.13 mmol) and TFA (0.30 mL, 4.0 mmol). The reaction mixture was allowed to warm to room temperature and stirred for 1.5 h before being concentrated and chromatographed (EtOAc) to provide a clear oil (15 mg, 0.03 mmol). The reaction was quenched with a saturated NaHCO_3_ solution (10 mL) and separated. The aqueous phase was extracted with DCM (10 mL × 3), and the combined layers were washed with brine, dried over Na_2_SO_4_, and filtered. After removal of the solvent, the residue was purified by flash chromatography on silica gel, eluting with DCM/EA (1/1) to afford **15a** in 67% yield as a white foamy solid. *R*_f_ = 0.42 (PE/EA 2/3). ^1^H-NMR (600 MHz, CDCl_3_): δ 7.78 (s, 1H), 7.14 (d, *J* = 9.5 Hz, 1H), 6.51 (d, *J* = 7.3 Hz, 1H), 5.67 (dd, *J* = 20.2, 11.0 Hz, 1H), 5.32 (dd, *J* = 17.5, 9.7 Hz, 2H), 5.15 (dt, *J* = 36.5, 7.5 Hz, 1H), 4.65 (dd, *J* = 9.4, 3.2 Hz, 1H), 4.26 (dd, *J* = 17.5, 2.8 Hz, 1H), 4.05 (d, *J* = 11.4 Hz, 1H), 3.30 (d, *J* = 11.4 Hz, 1H), 3.17 (m, 2H), 2.76 (d, *J* = 16.4 Hz, 1H), 2.56 (m, 3H), 2.42 (m, 2H), 2.15 (m, 5H), 1.87 (s, 3H), 0.69 (d, *J* = 6.9 Hz, 3H), 0.50 (d, *J* = 6.8 Hz, 3H). ESI-MS (*m/z*): 515.2 [M + H]^+^. HRMS-ESI (*m/z*): [M + H]^+^ calcd. for C_21_H_27_FN_4_O_4_S_3_H: 515.1251, found: 515.1252.

For **15b**: 59%, *R*_f_ = 0.42 (PE/ EA 2/3). ^1^H-NMR (600 MHz, CDCl_3_) δ 7.66 (s, 1H), 7.19 (m, 2H), 6.84 (m, 5H), 6.15 (s, 1H), 5.78 (dd, *J* = 17.9, 9.4 Hz, 1H), 5.16 (dt, *J* = 36.3, 7.5 Hz, 1H), 5.04 (dd, *J* = 17.4, 8.2 Hz, 1H), 4.93 (m, 1H), 4.27 (d, *J* = 17.1 Hz, 1H), 4.11 (d, *J* = 11.2 Hz, 1H), 3.26 (d, *J* = 11.3 Hz, 1H), 3.22 (dd, *J* = 14.0, 3.0 Hz, 1H), 3.12–3.08 (m, 1H), 3.07 (d, *J* = 5.9 Hz, 1H), 2.67 (d, *J* = 14.8 Hz, 1H), 2.54 (dd, *J* = 14.4, 7.1 Hz, 1H), 2.40 (m, 2H), 1.83 (s, 3H). ESI-MS (*m/z*): 563.0 [M + H]^+^. HRMS-ESI (*m/z*): [M + H]^+^ calcd. for C_25_H_27_FN_4_O_4_S_3_H: 563.1251, found: 563.1254.


*S-((Z)-4-Fluoro-4-((5R,8S,11S/R)-8-isopropyl-5-methyl-6,9,13-trioxo-10-oxa-3,17-dithia-7,14,19,20-tetraazatricyclo[14.2.1.12,5]icosa-1(18),2(20),16(19)-trien-11-yl)but-3-en-1-yl) octanethioate (*
**16a**
*)*


The free thiol **15a** (26 mg) was dissolved in dry DCM (10 mL) and cooled to 0 °C. The mixture was successively treated with Et_3_N (14 µL, 0.1 mmol) and octanoyl chloride (42 µL, 0.25 mmol). The reaction was allowed to warm to rt and stirred for 2 h, and then quenched with a saturated NaHCO_3_ solution (10 mL) and separated. The aqueous phase was extracted with DCM (10 mL × 3), and the combined layers were washed successively with brine, dried over Na_2_SO_4_, and filtered. After removal of the solvent, the residue was purified by flash chromatography on silica gel, eluting with DCM/EA (1/1) to afford **16a** (61%) as a white foamy solid. *R*_f_ = 0.42 (DCM/EA 3/1). [α]^20^_D_: +34.3 (*c* 0.3, CHCl_3_). ^1^H-NMR (600 MHz, CDCl_3_): δ 7.78 (s, 1H), 7.13 (d, *J* = 9.5 Hz, 1H), 6.42 (d, *J* = 7.8 Hz, 1H), 5.65 (m, 1H), 5.32 (dd, *J* = 17.5, 9.8 Hz, 1H), 5.10 (dt, *J* = 7.2, 2 Hz, 1H), 4.65 (dd, *J* = 9.3, 2.9 Hz, 1H), 4.26 (dd, *J* = 17.5, 2.8 Hz, 1H), 4.05 (d, *J* = 11.3 Hz, 1H), 3.28 (d, *J* = 11.3 Hz, 1H), 3.16 (dd, *J* = 16.4, 11.4 Hz, 1H), 2.90 (t, *J* = 7.1 Hz, 2H), 2.74 (d, *J* = 14.8 Hz, 1H), 2.54 (t, *J* = 7.5 Hz, 2H), 2.37 (m, 2H), 2.13 (m, 1H), 1.88 (s, 3H), 1.65 (m, 2H), 1.28 (m, 8H), 0.88 (m, 3H), 0.68 (d, *J* = 6.7 Hz, 3H), 0.49 (d, *J* = 6.7 Hz, 3H). ^13^C-NMR (150 MHz, CDCl_3_): δ 199.35, 173.64, 168.96, 168.84, 167.90, 164.62, 155.25 (d, *J* = 256 Hz), 147.49, 124.41, 108.71 (d, *J* = 12.6 Hz), 84.40, 69.95 (d, *J* = 29.6 Hz), 57.54, 44.17, 43.35, 41.17, 37.47, 34.33, 31.63, 28.92, 27.9, 25.64, 24.13, 23.86, 22.61, 18.86, 16.51, 14.09. ^19^F NMR (376 MHz, CDCl_3_): δ −124.76 (dd, *J* = 35.9, 20.9 Hz). ESI-MS (*m/z*): 663.3 [M + Na]^+^. HRMS-ESI (*m/z*): [M + Na]^+^ calcd. for C_40_H_41_FN_4_O_4_S_3_Na: 663.2115, found: 663.2139.

For **16b**: 66%, *R*_f_ = 0.42 (DCM/EA 3/1). [α]^20^_D_: +36.6 (*c* 0.025, CHCl_3_). ^1^H-NMR (150 MHz, CDCl_3_) δ 7.64 (s, 1H), 7.14 (d, *J* = 7.7 Hz, 2H), 7.00–6.74 (m, 5H), 6.15 (d, *J* = 8.0 Hz, 1H), 5.76 (dd, *J* = 19.2, 9.9 Hz, 1H), 5.12 (dt, *J* = 36.3Hz, 7.5 Hz, 1H), 5.06 (d, *J* = 9.0 Hz, 1H), 4.92 (t, *J* = 8.1 Hz, 1H), 4.92 (t, *J* = 8.1 Hz, 1H), 4.24 (dd, *J* = 17.4, 2.1 Hz, 1H), 4.10 (d, *J* = 11.2 Hz, 1H), 3.23 (d, *J* = 11.2 Hz, 1H), 3.21 (d, *J* = 2.8 Hz, 1H), 3.06 (m, 2H), 2.88 (t, *J* = 7.1 Hz, 2H), 2.66 (d, *J* = 16.0 Hz, 1H), 2.53 (t, *J* = 7.5 Hz, 2H), 2.36 (dd, *J* = 14.4, 7.2 Hz, 2H), 1.83 (s, 3H), 1.64 (dd, *J* = 14.0, 7.0 Hz, 3H), 1.27 (m, 8H), 0.89 (dd, *J* = 9.0, 4.6 Hz, 3H). ^13^C-NMR (150 MHz, CDCl_3_): δ 198.61, 173.12, 168.27, 167.47, 166.35, 163.04, 146.63 (d, *J* = 28.5 Hz), 134.52, 129.03, 127.22, 125.29, 123.03, 107.82 (d, *J* = 23.4 Hz), 83.63, 69.52 (d, *J* = 30.9 Hz), 53.57, 43.52, 41.84, 40.37, 37.00 (d, *J* = 18.3 Hz), 30.98, 29.07, 28.27, 27.27, 24.99, 24.35, 23.20, 21.95, 13.45 (d, *J* = 8.6 Hz). ^19^F-NMR (376 MHz, CDCl_3_): δ −124.76 (dd, *J* = 36.1, 19.3 Hz). ESI-MS (*m/z*): 689.2 [M + H]^+^. HRMS-ESI (*m/z*): [M + H]^+^ calcd. for C_33_H_41_FN_4_O_5_S_3_: 689.2296, found: 689.2302.

### 3.2. Biological Evaluation

#### 3.2.1. Recombinant Human HDAC1, HDAC2, HDAC3, HDAC6, and HDAC8 Enzymatic Assays

The assays were carried out by Shanghai ChemPartner Co., Ltd. (Shanghai, China). Briefly, different concentrations of compounds were incubated with recombinant human HDAC1, HDAC2, HDAC3, HDAC6, and HDAC8 (BPS Biosciences, San Diego, CA, USA) at room temperature for 15 min, which was followed by adding Ac-peptide-AMC substrates to initiate the reaction in Tris-based assay buffer. Reaction mixtures were incubated at room temperature for 60 min in the HDAC1, HDAC2, HDAC3, and HDAC6 assays, and were incubated for 240 min in the HDAC8 assay. Then a stop solution containing trypsin was added. The coupled reaction was incubated for another 90 min at 37 °C. Fluorescent AMC released from substrate was measured in SynergyMx (BioTek, Winooski, VT, USA) using filter sets as excitation = 355 nm and emission = 460 nm. IC_50_ values were calculated by GraphPad Prism version 4.00 Windows (GraphPad Software, San Diego, CA, USA).

#### 3.2.2. Cytotoxic Effect Tested by 3-(4,5-Dimethylthiazol-2-yl)-2,5-Diphenyltetrazolium Bromide (MTT) Assay

The MTT assay was used to determine the cytotoxic effect of these compounds on A549, HCT116, MDA-MB-231, and SK-OV-3 tumor cells. Cells (600,000 cells/well) were seeded in 96-well plates and incubated for 24 h before being treated with various concentrations of compounds or solvent control. Cells were further incubated for 72 h and then treated with MTT and incubated for another 3 h. The media were then removed and 100 µL DMSO was added to each well. The absorbance at 550 nm was measured by a SpectraMAX340 microplate reader (Molecular Devices, Sunnyvale, CA, USA) with a reference wavelength at 690 nm. Largazole was used as a positive control in the assay.

## 4. Conclusions

Largazole and Psammaplin A, two marine natural products belonging to a family of sulfur-containing natural products with potent HDAC inhibitory activities, contain the similar side chain unit. Inspired by the structural similarity, we hypothesized the fluoro olefin moiety would mimic both the alkene moiety in Largazole and amide moiety in Psammaplin A, and thus designed and synthesized two novel fluoro olefin analogs of Largazole. The preliminary biological assays showed that the fluoro analogs possessed comparable Class I HDAC inhibitory activities, indicating that this kind of modification on the side chain of Largazole was tolerable.

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
