# Peer review of "Synthesis and Preliminary Biological Evaluation of Two Fluoroolefin Analogs of Largazole Inspired by the Structural Similarity of the Side Chain Unit in Psammaplin A"

_marinedrugs, 2019, doi:10.3390/md17060333_

Round 1

Reviewer 1 Report

In the manuscriptSynthesis and preliminary biological evaluation of two fluoroolefin analogs of Largazole inspired by the structural similarity of the side chain unit in Psammaplin A, the authors set out to study the efficacy of analogs of Largazole, a histone deacetylase inhibitor.  The results reported here suggest the potential usefulness of these analogs in inhibiting Class I HDACs.  This is a revision of a previously reviewed manuscript.

The major concerns were largely met by the author from the last submission. The IC50 assays and growth assays are now included, as requested, and provide the appropriate context for the table summaries. Editing has improved the clarity of the paper, although some grammatical issues remain (a partial list is included below).  Still, with minor editing, this paper would be appropriate for publication.

Minor concerns:

- Table 2 now contains a graph of data and is now more appropriately labeled as a figure.

Incomplete list of editing mistake:

Line 20: Fragment, not fragement

Line 117: Assimilar

Line 119:  and almost thesame

Line 123: with anamide bond or aryl ring

Line 411: should read “(MTT) assay

Line 412: cytotoxic, not cytotxic

Author Response

Reviewer 1:

Editing has improved the clarity of the paper, although some grammatical issues remain (a partial list is included below).  Still, with minor editing, this paper would be appropriate for publication.

Minor concerns:

- Table 2 now contains a graph of data and is now more appropriately labeled as a figure. 

Incomplete list of editing mistake:

 Line 20: Fragment, not fragement

 Line 117: Assimilar

 Line 119:  and almost thesame

 Line 123: with anamide bond or aryl ring

 Line 411: should read “(MTT) assay

 Line 412: cytotoxic, not cytotxic

Response:

1. As suggested, we have added Figure 3, which appeared in the Table 2 of the previous revised version. Now Table 2 doesn't contain a graph of data.

 2. As suggested, we have corrected the editing mistakes occurring in Line 20, Line 117, Line 119, and Line 123, Line 411 and line 412, and highlighted them in red color.

3. As suggested, we asked for another English-editing expert to examine our manuscript again,  corrected the mistakes and highlighted them in red color.

Reviewer 2 Report

Dear authors,

I reviewed the revised version of the manuscript. I can see all the efforts that authors made to address the comments and and feel the manuscript is ready for acceptance.

Thank you.

Author Response

Thanks for the comments. As suggested, we asked for another English-editing expert to examine our manuscript again, and corrected the mistakes and highlighted them in red color.

This manuscript is a resubmission of an earlier submission. The following is a list of the peer review reports and author responses from that submission.

Round 1

Reviewer 1 Report

In the manuscript Synthesis and preliminary biological evaluation of two fluoroolefin analogs of Largazole inspired by the structural similarity of the side chain unit in Psammaplin A, the authors set out to study the efficacy of analogs of Largazole, a histone deacetylase inhibitor.  The results reported here suggest the potential usefulness of these analogs in inhibiting Class I HDACs.

While the topic of this paper is certainly of interest to the field, there are several key issues with the manuscript that would prevent me from recommending its publication.  First and foremost is the lack of the primary data for each experiment: we are given a table summary of two key experiments, but no graphs to see the IC50 experiments themselves.  It’s important that we see this data, including error bars for each data point, to make determinations of the reliability of these experiments.  A table summary, while useful, is wholly unconvincing on its own.

The second issue has to do with editing and grammar of the manuscript.  There are multiple instances throughout the paper where incorrect grammar and sentence structure are used.  This significantly detracts from the quality of the paper.

While I would not recommend rejecting this paper outright, without the above changes, it is difficult to assess this paper. However, with editing and the inclusion of primary data, this paper could be of significant interest, and thus it is my suggestion that this paper be allowed to be reconsidered after major revision.

Specific concerns are outlined below

Major concerns:

1) There are multiple grammar / word choice errors throughout.  It would be impossible to highlight them all.  Major editing is needed.  However, a few such instances are highlighted below:

Line 112: Most of these results were consistent well with the reported results

Line 118 : Alike the parent free thiol

Line 120: and could kept almost same selectivity over Class II HDACs 

2) Some kind of primary data for the HDAC activity assays is necessary (either in the main figures or the supplement).  Since IC50s are being reported on, it would be necessary to see the data these are based on to judge the accuracy and reliability of this data.

3) The same is true with the cell growth data:  Again, the IC50 growth curve would be appropriate.  Adriamycin is mentioned as a positive control, but this doesn’t mean much if we can’t see the data.  In addition, the cell assay is not thoroughly explained.  For example, the procedure uses MTT for a colorimetric assay, but the abbreviation and purpose of MTT use are never clarified.  

Reviewer 2 Report

Dear authors, 

I reviewed the manuscript entitled "Synthesis and preliminary biological evaluation of two fluoroolefin analogs of Largazole inspired by the structural similarity of the side chain unit in Psammaplin A". The manuscript adds into the pool of information in understanding histone deacetylase inhibitors and finding the synthetic analog of the HDACs. In the manuscript, the authors reports synthesis of fluoro analogs of Largazole and the preliminary study of their effects using enzymatic and cellular assays. Though the authors have performed important study using combination of synthetic chemistry approach and biological assays, I have few of the comments below to add clarity to the works mentioned in the manuscript. My comments are:

1) Authors have mentioned about the unsuccessful reports in the last sentence of the Abstract and Conclusions sections "....the previous unsuccessful reports." Were they published reports and cited in the manuscript? Please cite the reports if published or mention elsewhere in the manuscript in the relevant section especially, in the Results and Discussion section.

2) I strongly recommend to include dose-response curve for the inhibition assays. The author mentions 10-dose IC50 with threefold dilution in duplicate. It would be helpful for the interested community to see the dose-response curve with the points indicating inhibition activity at different time point with error bar. I do not see error bar for the IC50 value in the submitted tables.

3) Were the cellular assays performed using 15a and 15 b compounds? Please report the IC50 for those compounds in Table 2.

4) The NMR spectra submitted as supplementary materials need to be organized by providing the titles and subtitles for respective images. I can see subtitles and NMR spectra but the organization is confusing. Please arrange those spectra with numbers to avoid confusion.

Minor comments:

1) In line# 401, the volume needs correction. The segment “…removed and 100 mL DMSO was added….” mentions the volume that does not seem correct.

2) Please correct Bruk as Bruker Instruments (line # 165).

3) Mention extended form of HRMS.

4) If possible and space limitation permits, I would suggest authors to submit mass spectrum (MS and MS/MS) for the synthesized compounds.

Thank you.